# A Kaleidoscope of Keratin Gene Expression and the Mosaic of Its Regulatory Mechanisms

**DOI:** 10.3390/ijms24065603

**Published:** 2023-03-15

**Authors:** Ekaterina P. Kalabusheva, Anastasia S. Shtompel, Alexandra L. Rippa, Sergey V. Ulianov, Sergey V. Razin, Ekaterina A. Vorotelyak

**Affiliations:** 1Cell Biology Laboratory, Koltzov Institute of Developmental Biology, Russian Academy of Sciences, 119334 Moscow, Russia; 2Department of Molecular Biology, Faculty of Biology, M.V. Lomonosov Moscow State University, 119234 Moscow, Russia; 3Laboratory of Structural-Functional Organization of Chromosomes, Institute of Gene Biology, Russian Academy of Sciences, 119334 Moscow, Russia

**Keywords:** keratin genes, epithelium, enhancer, transcription regulation, chromatin spatial organization

## Abstract

Keratins are a family of intermediate filament-forming proteins highly specific to epithelial cells. A combination of expressed keratin genes is a defining property of the epithelium belonging to a certain type, organ/tissue, cell differentiation potential, and at normal or pathological conditions. In a variety of processes such as differentiation and maturation, as well as during acute or chronic injury and malignant transformation, keratin expression undergoes switching: an initial keratin profile changes accordingly to changed cell functions and location within a tissue as well as other parameters of cellular phenotype and physiology. Tight control of keratin expression implies the presence of complex regulatory landscapes within the keratin gene loci. Here, we highlight patterns of keratin expression in different biological conditions and summarize disparate data on mechanisms controlling keratin expression at the level of genomic regulatory elements, transcription factors (TFs), and chromatin spatial structure.

## 1. Introduction

The human genome contains a total of 54 functional keratin genes that code 28 keratins of type I and 26 keratins of type II. Type I keratins include 17 epithelial (K9-28) and 11 hair-specific keratins (K31-40), and type II keratins include twenty epithelial (K1-8 and K71-80) and six hair-specific keratins (K81-86) [1]. Keratin filaments are obligate heterodimers of type I and type II chains in a 1:1 molar ratio. Usually, keratin expression is described as paired due to the conserved composition of keratin heterodimers: 8/18, 5/14, 4/13, 1/10, etc.

The structure and functions of an epithelium depend on its mechanical barrier properties. The type of epithelium gradually changes from a simple single layer in the internal organ epithelia to a keratinized stratified epithelium in the skin epidermis. The keratin expression profile changes respectively: keratins 8/18, 7, 19, and 20 are typical for simple epithelium; expression of keratins 5/14 and 15 arises in pseudostratified epithelium; and stratified epithelia are positive for keratins 4/13 in mucous, keratins 1/10, 2 and 9 in the skin epidermis, and keratins 3/12 in the cornea. The widest spectrum of keratin expression is inherent to the hair follicle and the nail bed. The expression pattern of keratin has been well described and characterized [2,3,4]. Dysfunctions of tissues and organs due to violations of keratin distribution could be associated with impaired control of the keratin expression or genetic abnormalities. Since the expression pattern of keratins is very similar between humans and mice, a lot of human diseases have been modeled using mice knockouts. However, it should be taken into account that mice lack several orthologs of human keratins, and the expression pattern is also slightly different for several keratins [5].

Genes encoding human type I and II keratins are clustered into two distinct chromosomal regions:17q12–q21 for type I keratins (except K18) and 12q11–q13 for all type II keratins and K18 (Figure 1) [2]. The genes of conserved keratin pairs are located in different chromosomes, except keratins 8/18. Clustered localization is a conserved feature of keratin genes in terrestrial vertebrates [6] and suggests the presence of mechanisms for the coordinated keratin gene expression within clusters and on different chromosomes. In higher eukaryotes, gathering paralogous genes into a common cluster usually reflects the necessity to choose one or several of these genes to be expressed whereas keeping the other genes silent [7]. As shown in studies of several multigenic loci, i.e., protocadherin, immunoglobulin, Hox, and globin gene clusters, reconfiguration of spatial regulatory landscape within gene loci is required for the proper regulation of gene activity during differentiation and development [8,9,10,11]. In this review, we summarize the present data on the keratin gene-related transcription factors (TFs) and keratin-specific enhancers regulating keratin transcription and its switching in different biological conditions. In particular, we discuss a possible impact of a large-scale chromatin spatial structure and distant contacts between keratin regulatory elements in coordination with keratin expression.

## 2. Keratin Expression Pattern

The principal characteristic of tissue-specific keratin expression pattern is the transcription of conserved keratin pairs: keratin pair 8/18 is typical for simple epithelia, 5/14—for basal layer in pseudostratified and stratified epithelia, 6/16—for regenerating tissue, etc. However, the veritable paired expression occurs quite rarely. More often, the expression pattern of one of the paired keratins is much more widespread than that of its heterodimer partner [12,13,14]. Even when both keratin heterodimers are present in the same cell, the expression at the mRNA level may have a more significant correlation with additional keratin rather than paired keratin [15]. It is thus more correct to describe the cell type- and tissue-specific patterns of keratins as a complex consisting of a key pair of keratins and additional keratins (Table 1).

Simple epithelia keratin complex includes the keratin pair 8/18 and could be complemented with additional keratins 7, 19, 20, and 23. The first keratin 8/18 filaments are detected in a subset of cells of the eight-cell mouse embryo; their expression contributes to the isolation of the trophectoderm layer [16]. The blastocyst stage keratin expression pattern is completed by keratins 7 and 19 [145], whereas the inner cell mass is devoid of keratins [16]. In adult parenchymatous organs, the spectra of additional keratins become more complex along with an increase in the mechanical stress within an epithelium. For instance, hepatocytes of liver parenchyma are positive only for keratins 8/18, while the bile ducts also express keratins 7 and 19 [21,22]. A selective expression of keratin 23 is detected in the mouse gallbladder and common bile duct [23]. The expression of additional keratins appears de novo in pathological conditions [23,146]. In several cases, the keratin complex could lack the expression of paired keratins 8/18. Endothelia of normal human veins, venules, and lymphatics commonly exhibited focal positivity for keratin 7 and 18, whereas keratin 8 was not detected in non-neoplastic endothelia [46,47], however, most often, the loss of keratin expression indicates an epithelial-mesenchymal transition during carcinogenesis [147]. Keratins of simple epithelia are also expressed in non-epithelial tissues: smooth [47,148], skeletal [149,150], and cardiac muscle [151].

Pseudostratified epithelium combines the features of both simple and stratified-epithelia. A pair of keratins 8/18 (typical for simple epithelia) is expressed throughout the entire bulk of the pseudostratified epithelium or tends to shift to the outer layers [49,59,64]. The first keratins typical for the stratified epithelium appear in the basal layer. This probably provides increased mechanical strength of a cell mass as compared to simple epithelia. Keratin 5 is identified in the basal layer [43,50,59,64], while only a minor cell subpopulation is positive for its typical counterpart, keratin 14. Keratin 14 expression increases during injury-induced regeneration [13,51,152]. The keratins of pseudostratified epithelia could include keratins typical for other keratin complexes such as keratin 4/13 of suprabasal layers of stratified epithelia or stress-induced keratins [59].

Stratified epithelia are characterized by two complexes of keratins. Basal layers are positive for keratins 5/14 pair and additional keratin 15 [70,95,153]. The second keratin complex is expressed in the differentiating post-mitotic layers and differs depending on the epithelial localization. Mucosal epithelia express keratin pair 4/13 [154,155], cornifying epithelia are positive for keratins 1/10 with additional keratins 2 and 9 [15], keratin 12 and 3 (the last is human-specific in this complex) are expressed in specialized epithelial cells of the cornea [84].

Wound healing or other pathological conditions activate the expression of the stress-induced complex: keratins 6A, B, and C form the pair with keratins 16 and 17 [95]. These keratins promote cell migration, proliferation, and survival [156,157,158]. The stress-induced complex of keratins is associated with the downregulation of keratin 15 [159,160], keratins 1/10 [161,162], and keratins 4/13 [159], thereby, epithelia-specific keratins are replaced by the stress-induced keratins. Several wound-healing keratins expressed in healthy tissues are related to high proliferation and migration levels in hair follicles [118] and nails [129,163], or high mechanical stress in palmoplantar skin [164].

The hair keratin complex possesses the widest spectrum and the most restricted localization (Figure 2). Outer root sheath (ORS), the outermost compartment which is continuous with the interfollicular epidermis, expresses keratins 5/14, 15, 6/16, and 17 [165]. A specialized part of the follicle ORS known as the bulge contains hair follicle stem cells [166,167]. During the active growth phase, anagen, bulge cells proliferate, producing progenitor cells migrating down the hair bulb at the base of the hair follicle, where they actively proliferate and generate the hair matrix. Matrix cells move upward and differentiate, developing into an inner root sheath (IRS) consisting of Henle, Huxley, and cuticle layers, as well as a hair shaft consisting of cuticle, cortex, and medulla. The keratins expressed in the IRS include 25–28 of type I and 71–74 of type II. For the hair shaft, 31–40 of type I and 81–86 of type II are typical [2,168,169]. A companion layer derived from matrix cells separates the IRS and hair shaft from the ORS. It expresses non-hair keratins: 6,16, 17, and 75 [170]. The hair medulla exists only in certain types of hair follicles: terminal in humans and guard in mice, respectively. This hair shaft central part possesses the most complicated spectra of keratins, including keratins of ORS, IRS, hair shaft cuticle, and cortex (Figure 2) [168]. Several noncanonical skin or hair keratins could be identified in hair follicles of different mammalian taxons, in particular, including a unique mammal-specific splice variant of keratin 80 in humans [171] or mucosa-related keratin 4 and 13 in cashmere goats [172]. Hair and nails, as well as other hard keratinized mammalian skin derivatives such as horns and hoofs, are examples of highly restricted unique keratin expression areas. Equine hoof lamellae epithelia, which are homologous of the nail bed, possess the expression of unique keratins 42 and 124. Murine keratin 42 expression has been localized to the nail matrix and nail bed, while the rodent orthologs of keratin 124, keratin 90, have not been characterized beyond genomic mapping and identification as likely functional genes in those species. In the human genome, K42 and K124 exist only as pseudogenes [173].

Gene targeting in a mouse model indicates the leading role of type II keratins in the organization of the above-mentioned complexes. During mouse embryogenesis, the expression of type II keratins usually precedes the expression of their type I counterparts [145]. Deletion of the entire keratin type II locus leads to the prenatal death of mouse embryos at E9.5 caused by a sequence of keratin-dependent mechanisms that leads to metabolic failure in embryonic and extraembryonic tissues [174]. Keratins 8 and 18 are first expressed in embryogenesis. Type II keratin 8 knockout results in lethality at E12.5 and complete embryos resorption at E15.5 [175], while type I keratin 18 null mice show no fetal lethality and have a normal lifespan with only minor liver abnormalities due to replacement by another type I keratin 19 [176]. Keratin 8 knockout suppresses the production of keratins 18, 19, and 20 at the protein level but not at the RNA level in the pancreas and intestines [28,177,178]. Although the expression of keratin 7 decreased, its level was still sufficient for the formation of a low amount of 7/18 and 7/19 keratin heterodimers [177]. Keratin 5 knockout induces neonatal death, while the lack of keratin 14 compensates for keratin 15 in the skin epidermis [179,180]. The importance of type II keratins is confirmed by the fact that keratin 5 is often present in the basal layer of pseudostratified epithelium, while keratin-14 expression is sparse [43,50,52,59,64]. However, recently published details about mouse epidermal keratinocytes differentiation revealed the expression of keratin 10 belonging to type I of keratins prior to type II keratin 1 at a protein level [181]. These novel results indicate the importance of further investigation of the transcription activation of paired keratins within the keratin loci for the identification of the basic principles of their coordinated expression.

**Figure 2 ijms-24-05603-f002:**
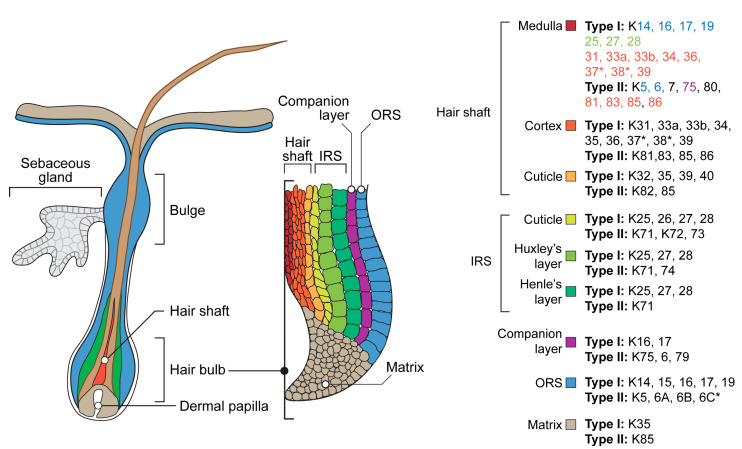
Schematic representation of the hair follicle and a list of keratins expressed in its internal structures. *—absent in the mouse genome. ORS—outer root sheath, IRS—inner root sheath. Note that keratins expressed in the medulla are colored according to their presence in other structures of the hair follicle [2,153,168,182,183,184].

## 3. Regulation of Keratin Gene Transcription

In higher eukaryotes, enhancer elements containing clustered binding sites for TFs are the main regulators of tissue-specific gene expression. Within the keratin gene loci, early studies were aimed at detecting local regulatory elements that provide effective activation of keratin gene expression. Two DNase I hypersensitive sites (HSII and HSIII) located upstream of the transcription start site (TSS) of the K14 gene were found to be jointly able to efficiently activate tissue-specific transcription of the reporter gene in transfected primary keratinocytes and transgenic mice [185,186]. Interestingly, each of these HSs alone was unable to drive tissue-specific gene expression in transgenic mice: HSII weakly activated transcription of a reporter gene in epidermal and mesenchymal cells [186], whereas HSIII drove expression in IRS cells where the K14 gene is normally inactive. This finding suggests cooperativity in the functioning of these HSs as regulators of K14 expression [187].

Similar studies using reporter constructs were carried out for other keratin genes. Enhancer elements were detected upstream from the K5 gene TSS [188] and the K6a gene [189] and within the 3′-flanking region of the K1 gene [190]. The enhancer of the K19 gene was found at a significant distance downstream from the TSS, indicating DNA looping as a potential mechanism of the enhancer-promoter contact [191]. A noteworthy example of the gene regulatory landscape is described for the K18 gene. Initially, it has been shown that a 10 kb fragment including the K18 gene is sufficient to drive tissue-specific transcription indicating that it contains all necessary regulatory information. Later, 5′- and 3′-flanking regions of this 10 kb fragment were found to ensure copy number-dependent and position-independent transcription via insulating the K18 gene from the local neighborhood [192]. Furthermore, several local enhancers have been found in this locus, including the major enhancer element located in the K18 first intron and capable of driving transcription solely [193]. This enhancer contains binding sites for AP1 and Ets TFs, jointly promoting transcription activation of the K18 gene during tumorigenesis and differentiation [194,195]. Interestingly, the first intron enhancer, minimal promoter, and 5′-flanking region of the K18 gene were used in the expression cassette to drive tissue-specific expression of the CFTR gene. This expression cassette can potentially be used for the development of gene therapy strategies for cystic fibrosis [54]. Local enhancers were also found upstream of the TSS and within the sixth exon of the K18 gene [27,196]. Similarly, local control elements are located inside the K8 coding region [197]. This complex regulatory landscape is potentially important for the K8/18 regulation during early development since this pair represents the first keratins expressed in an embryo.

The expression of several keratin genes in each epithelial cell type suggests the presence of mechanisms ensuring coordinated regulation of transcription activation/switching in keratin gene loci. One possibility is regulation by a limited set of tissue-specific and common TFs cooperatively binding to keratin gene promoters and distant regulatory regions. While a number of TFs were identified as keratin regulators in studies of individual keratin genes (Table 1), only a few TFs play the role of master-regulators of keratin loci. Specifically, the p53-related protein p63 is one of the most significant TFs controlling keratin expression [198]. p63 is present in keratin 5-positive cells of lung airway epithelia [43,50,52], esophagus [70], mammary glands [64], and skin epidermis [96,97,199]. The p63 gene is transcribed from two alternative promoters, generating several isoforms of TAp63 transcripts from an upstream promoter and ΔNp63 variants from the downstream promoter located within the intron 3. The abnormal skin phenotype observed in genetically modified mice has revealed that ΔNp63 initiates the process of differentiation and maintains the proliferation of basal layer keratinocytes, while TAp63 drives the expression of proteins specific to the upper layer of the epidermis, such as Ets-1, keratins 1/10, profilaggrin, involucrin and transglutaminase type 3 and 5 [96,199]. p63 elicits chromatin remodeling and increases chromatin accessibility at epidermal enhancers [200,201,202]. Interestingly, p63 also modulates the expression of miRNAs involved in the regulation of keratin gene transcription (for the review of the miRNA role in keratin biology, see [203]).

Other master regulators of keratin expression are Kruppel-like factors, in particular, KLF4 involved in keratin expression switching during differentiation in stratified epithelia. Klf4 knockout attenuates cells differentiation as well as keratins 4/13 expression in the esophagus [71], downregulation of keratin 12 expression in the cornea [85], and impaired skin barrier formation [98], which is also associated with declining expression of late epidermal differentiation proteins filaggrin and loricrin [99]. KLF4 and KLF5 play key roles in the proliferation and differentiation of esophagus epithelia. Klf5 maintains the low-differentiated state of epithelial cells, while Klf4 inhibits the Klf5 in progenitor cells to induce differentiation [29,71]. Microarray analysis has shown that many keratin genes were upregulated after KLF4 induction, indicating its role in epithelial differentiation [100]. Klf4 acts as a transcriptional activator of epithelial genes and as a repressor of mesenchymal genes [204] and is also able to directly bind promotor regions of several keratin genes.

Together, p63 and KLF4 are cornerstones of keratin expression regulation and keratinocyte phenotype determination since their ectopic expression induces the conversion of fibroblasts into keratinocyte-like cells by activating genes crucial for epithelial lineage specification [101,198].

In certain conditions, the expression of keratins typical for differentiated cells is repressed or markedly modulated. During wound healing, SNAI2 serves as a negative regulator of differentiation-related keratins, which probably promotes the activation of stress-induced keratins. SNAI2 is a C2H2-type zinc finger TF involved in epithelial-mesenchymal transition. Ectopic expression of SNAI2 downregulates keratin 10 as well as other differentiation markers, such as keratin 1, filaggrin, transglutaminase 1, SPRR1A, GRHL3, and KLF4 [205]. Although the expression of keratins 6, 10, and 14 are not altered in Snai2-deficient mice, the expression of keratin 8 is upregulated [206]. SNAI2 also inhibits the keratin 8/18 expression in breast cancer cells via binding to their promoter regions [207].

The hair follicle is characterized by the widest pattern of keratin expression (Table 1). The development and regeneration of the hair follicle are controlled by WNT, TGFβ, BMP, SHH, NOTCH, Eda/EdaR, etc. [208]. Lef1, a target of the WNT/β-catenin pathway, binds cooperatively with TFs Sp1, AP2-like, and NF1-like to activate hair follicle-specific genes, in particular, keratins [119]. Lef1 knockdown leads to sparse hair, with a complete loss of mouse whisker follicles [209]. Lef1 transactivates the expression of hair-specific keratins by interaction with β-catenin and presumably some other factors involved in promoting the hair cell fate [120]. Transient β-catenin stabilization may be a key event in the long-sought epidermal signal leading to hair development and implicating aberrant β-catenin activation in hair tumors [121]. WNT/β-catenin/Lef1 pathway activation is certainly one of the stages of reprogramming for interfollicular epidermal keratinocytes in the follicular direction during co-cultivation with trichogenic mesenchyme [210].

While some TFs are important for chromatin remodeling at promoters and activation of enhancers, other TFs potentially serve as looping factors mediating spatial interactions between regulatory regions within keratin loci. For instance, KLF4 forms molecular condensates and thus is able to stabilize enhancer-promoter interactions [211]. p63, which extensively binds both keratin promoters and enhancers, is also involved in the stabilization of looping interactions in cooperation with the major architectural protein CTCF [212]. Thus, the spatial structure of chromatin should be considered as another level of regulation of keratin gene expression.

## 4. Chromatin Spatial Organization as a Regulator of Keratin Gene Expression

Prior to the discussion of the spatial structure of keratin loci, we briefly highlight key features of the genome spatial organization (Figure 3). In interphase, chromosomes occupy discrete non-random territories in nuclear space [213]. The specific feature of chromosome territories is a high frequency of intrachromosomal contacts compared to interchromosomal contacts, as has been shown both by chromosome conformation capture-based and FISH methods [214]. However, at least in the case of erythroid-specific genes [215] and clusters of olfactory receptor genes [216], and possibly in some other loci, the interchromosomal contacts are also functionally relevant.

Within chromosome territory, active and repressed loci are segregated into A and B compartments [214], which are subdivided into a spectrum of subcompartments distinguished by the patterns of histone marks, transcription activity, and replication timing. The results of recent studies suggested phase separation as a potential mechanism of chromatin compartmentalization [217]. A number of specific factors, including the master regulator of epidermal differentiation KLF4 [211], heterochromatin-associated protein HP1α, and components of transcription machinery [218], are involved in this process. However, other results suggest that chromatin masses are solid rather than liquid [219].

At the scale of 0.1–1 Mb, the genome is partitioned into topologically associated domains (TADs) [220,221] characterized by a high frequency of contacts between loci within the domain and depletion of interactions with the neighborhood. In mammals, TADs boundaries are demarcated by the presence of CCCTC-binding factor (CTCF) and cohesin complexes playing a crucial role in the loop extrusion serving as a mechanism of TADs and loop formation [220,222]. The loop extrusion model of TADs formation suggests that cohesin progressively extrudes chromatin fiber until it encounters the barrier, such as CTCF in convergent orientation [223] or the countermovement of transcription machinery that blocks cohesin traveling [222]. The importance of these factors is shown in numerous experiments using auxin-inducible degradation of CTCF [224] and components of the cohesin complex [225]. The majority of TAD boundaries are stable across different cell types and evolutionary conserved in related species [226], as well as coincide with replication domains [227]. TAD boundary disruption [228] and CRISPR/Cas9-mediated inversion [229] result in a reconfiguration of chromatin loop topology between enhancers and promoters and lead to gene misregulation. Thus, TADs could be considered structural and functional units of the genome.

Regulatory chromatin networks at various scales significantly change during cellular differentiation and the switching of gene expression programs. Chromosomes 12 and 17, which contain clusters of the keratin genes, extensively interact with each other at an early stage of keratinocyte differentiation [230]. This is potentially important for the coordinated expression of keratin genes from the two clusters to allow for the assembly of obligate heterodimers. In another study, global screening of regulatory regions at different stages of keratinocyte differentiation showed that the sets of enhancers and superenhancers are different and specific for a particular stage of differentiation [231]. In addition, the expression of keratin genes located on chromosome 12 and a group of epithelial-specific genes was found to be significantly upregulated by induced expression of KLF4 in human cells. Coregulation of keratin genes increases the possibility of the existence of a locus control region controlled by KLF4 [100].

Dynamics of the enhancer-promoter interactions within the keratin gene locus on chromosome 12 were revealed using the Capture-C method allowing the investigation of spatial contacts between specific regions of the genome [232]. Two classes of contacts between keratin promoters and enhancers were found: static, formed in progenitor cells, and dynamic, established during cell differentiation. Dynamic contacts are characterized by the acquisition of an active H3K27ac mark during differentiation and a significant increase in contact strength, whereas pre-established (static) contacts are associated with premarked H3K27ac. Remarkably, each sort of contact is regulated by a certain set of TFs. Stable contacts are preferably associated with cohesin complex and EHF. In contrast, dynamic interactions are depleted with cohesin and require KLF4 and ZNF750, which contribute to both enhancer activation and enhancer–promoter interactions. These contacts are established within several TADs whose boundaries are stable during epidermal differentiation. Similarly, the murine epidermal differentiation complex (EDC) locus, containing genes involved in the control of terminal epidermal differentiation, consists of several gene-rich and gene-poor TADs with stable boundaries [233]. However, in contrast to the keratin gene locus, a high frequency of enhancer-promoter contacts is observed not only within the encompassing TADs but also between gene-rich TADs. In this case, a putative superenhancer in one TAD activates target genes in the other TADs illustrating that, in at least some genome loci, TAD boundaries insulate large chromosome segments from each other but cannot prevent distant regulatory interactions. Recently, it has been shown that such distant contacts within mouse keratin locus are regulated by DNA deoxygenases Tet2/3 in a DNA methylation-dependent and DNA methylation-independent manner [234]. These data are of particular interest because they raise questions about the potential modulation of keratin gene expression by artificial recruitment of DNA-methyltransferases [235].

Finally, there is a noteworthy example of the role of the 3D genome organization in providing a regional specification of the skin appendages of birds. In addition to conventional α-keratin type I and II, birds and reptiles have a special class of intermediate filament proteins (β-keratins or corneous β-proteins, while structurally these proteins are not related to keratins), which serve as “building blocks” of various skin derivatives. Transcription activation of a proper type of the β-keratin genes is provided by two different epigenetic strategies. The first strategy is a differential activation of a specific subcluster of the β-keratin genes on chromosome 25 (e.g., feather keratin genes) regulated by a superenhancer, thereby providing regional specification and development of different skin appendages (scales, claws, feathers, etc.). Another strategy is the activation of a subset of individual keratin genes required for the formation of a particular type of feather. Specificity of interaction between H3K27ac-marked regions operating as loop anchors is mediated by TFs such as KLF4 and CTCF to create the region-specific interaction networks. Moreover, genes located in the same loop are coexpressed in the same skin region or development stage shown by immunocytochemical staining. Hence, while subclusters of keratin genes on chromosome 25 are regulated as single units, differential activation of a subset of feather keratin genes on chromosome 27 occurs by changing higher-order looping configurations [236].

## 5. Conclusions and Outlooks

Clustered localization of paralogous genes is a conserved feature of vertebrate genomes. Genes encoding keratins, as well as globins, protocadherins, immunoglobulins, olfactory receptors, KRAB-ZNF TFs, and ribonucleases of the A superfamily, are grouped into relatively compact genomic domains. These loci possess certain common properties, such as preferentially unidirectional transcription, activation, and repression of individual genes in different biological conditions and cell types, and the presence of complex regulatory elements [7]. Clustered localization implies tight control of the regulatory system within a domain: activation and repression on the principle of “all or nothing” is impossible, and thus “simple solutions” such as changes in the spectrum of expressed TFs are not applicable. Precise control of promoter activity could be implemented by reconfiguration of the locus spatial structure, which, in turn, requires modulation of the regulatory and architectural elements by tuning their epigenetic state with an impact from the master-regulators of keratin complexes. Hence, we suggest the following key questions to be addressed in future studies of keratin gene loci:(i)Reconstruction of the molecular cascades activating/switching keratin expression: from the incoming extracellular signal to activation of distinct keratin promoters.

The control of keratin expression provided by external signals from the epithelial microenvironment determines the cell stemness, the cell fate, the activation of regenerating behavior, and the regional specification of the epithelium. The plasticity of mammalian epithelia allows it to respond to stimuli from stromal and other niche cells by switching the differentiation program. Thus, epidermal keratinocytes are able to recapitulate the hair follicle [237,238] or palmo-plantar [239] skin differentiation program as well as corneal [240] and urethral [241] in tissue-specific substitutes with activation of specific keratin expression. At the same time, the key signals of newly formed epithelia-mesenchymal interactions remain unknown, as well as the intracellular pathways controlling the cell transdifferentiation, including the keratin complexes switching.

(ii)Disclosure of mechanisms preventing aberrant activation of keratin genes within complex regulatory landscapes of the keratin gene loci.

One of the markers of cancer progression and other pathologies is the appearance or upregulation of several keratin genes, including K17, K19, K6/16, etc. Keratins are not only routinely used for cancer diagnostics, but are also applicable for the prediction of invasiveness and aggressiveness of the tumor cell [242,243,244,245,246,247,248,249,250,251,252,253,254,255,256,257,258,259]. Following the aberrant expression, the reorganization of the keratin cytoskeleton affects the cell motility, survival, and resistance to external factors, which results in tumor treatment responsiveness [260,261,262,263,264,265,266,267,268,269]. The discovery of the molecular basis of aberrant keratin expression induction is one of the directions in cancer research. A potential role of oncogenetic TFs in the direct activation of individual keratin genes or possible reorganization of spatial contacts between keratin gene promoters and enhancers within keratin loci are questions to be addressed in future research.

(iii)Identification of mutations in regulatory regions associated with diseases caused by aberrant keratin expression.

Keratin gene mutations cause various hereditary diseases and keratinopathies, resulting in cellular and tissue defects [270,271,272,273,274,275,276,277,278]. Several mutations in different keratin genes manifest in very similar symptoms, thus causing different forms of the same disease. Epidermolysis bullosa simplex is a group of inherited disorders caused by mutations in at least seven different genes, including K5 and K14, characterized by recurrent blister formation as the result of skin and mucosa fragility [276,277,279,280]. Pachyonychia congenita, which could be a result of mutations in K6a, K6b, K6c, K16, or K17 genes, is associated with hypertrophic nail dystrophy and palmoplantar keratoderma [281,282]. Epidermolytic ichthyosis is caused by mutations in the genes K1, K2, and K10 [283,284,285]. Simple epithelia keratin mutations have been identified as risk factors for some intestine and liver diseases [12,286]. Many of these conditions can be reproduced for pathogenesis studies by genetically modifying mice [287]. The discovery of the genetic basis of keratin-associated disorders provides the basis for the development of reliable gene therapy strategies [288,289]. Recent evidence on the etiology and pathophysiology of inherited diseases revealed that not only mutations located in gene bodies but also in noncoding DNA could result in phenotypic abnormalities due to transcriptions disturbance [290,291,292,293,294,295,296,297]. Identification of the regulatory elements in keratin loci will provide new insights into congenital diseases driven by keratin cytoskeleton defects.

## Figures and Tables

**Figure 1 ijms-24-05603-f001:**
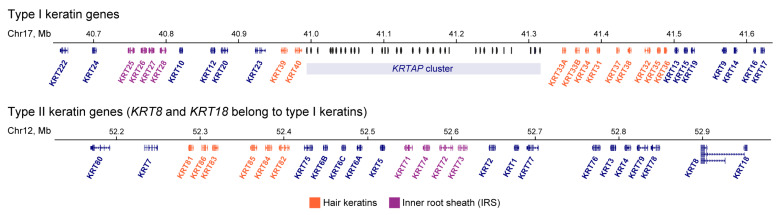
Loci of human keratin genes. Adopted from the UCSC genome browser. KRTAP—an array of genes encoding keratin-associated proteins.

**Figure 3 ijms-24-05603-f003:**
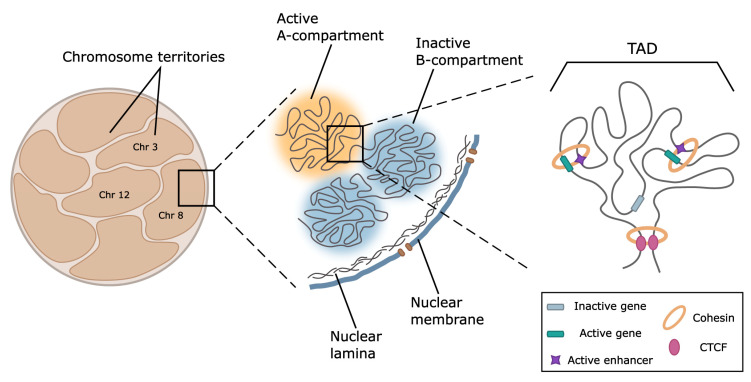
Schematic representation of the major levels of 3D genome organization. At the level of the whole nucleus and within chromosome territories, active and repressed genome loci are spatially segregated within A- and B-compartments. These structures are formed by large-distant contacts between regions possessing similar profiles of epigenetic marks. Regions forming inactive compartments tend to locate at the nucleus periphery, where they establish contact with the nuclear lamina. At the level of 100–1000 kb, chromosomes are folded into globular topologically associated domains (TADs). In mammals, TADs are predominantly formed by cohesin-driven chromatin extrusion, frequently terminating at CTCF-binding sites. In a number of well-characterized genome loci, TAD boundaries restrict the areas of enhancer action allowing regulatory systems to activate genes located within a TAD. According to the current paradigm, activation is provided by loop formation between the enhancer and cognate promoter.

**Table 1 ijms-24-05603-t001:** The pattern of keratin expression and TFss involved in keratin expression regulation. *—absent in the mouse genome. ORS—outer root sheath, IRS—inner root sheath.

Tissues (Organs) and Keratins	Transcription Factors	Links
**Simple Epithelia Keratin Complex**
**Trophectoderm:** *K*8/18, 7, 19	**CDX2, TEAD4, BAF155, AP-1, Ets**	[16,17,18,19,20]
**Liver:***Precursors: K*8/18, 19*Hepatocytes: K*8/18*Bile ducts: K*8/18, 7, 19, 23	**HNF-1, HNF-4alpha, HNF-3beta, C/EBPalpha, AP-1, SP1**	[21,22,23,24,25,26,27]
**Pancreas:***Acinar cells: K*8/18, 19, 20*Ducts: K*8/18, 7, 19, 20	**KLF4**, **KLF5**, **PDX1, MEIS1a, PBX1b, ELF3, Sp1, AP-2**	[28,29,30,31,32,33]
**Intestine:** *K*8/18, 7, 19, 20, 21	**CDX1, HNF-1α, HNF-4α, Cdx2, GATA-4** **AP1, SATB2**	[12,27,34,35,36,37,38,39,40]
**Kidney (Nephron):** *K*8/18, 7, 19	**HNF-4α, SIX1**	[36,41,42]
**Lungs (alveoli):** *K*8/18, 7, 19	**Nkx2-1**	[43,44,45]
**Endothelia:** *K*18, 7	**Hey2**	[46,47,48]
**Pseudostratified Epithelia Complex**
**Lungs:** *Bronchioles: K*8/18, 19, minor: *K*5, 6, 7, 15, 17*Trachea & bronchi:* *K*5, 15, 8/18, 6a, 19, minor: *K*14	**p63**, **SOX2**, **SOX21**, **KLF5**, **CFTR, ESE-2****FOXA1/A2**	[13,43,45,49,50,51,52,53,54,55,56,57,58]
**Urinary system:***Renal pelvis: K*8/18, 7, 19*Bladder: K*8/18, 5, 4, 7, 19, 20, 13	**p63**, **PPARγ1, GATA3, FOXA1**	[14,59,60,61,62,63]
**Mammary gland (ducts):** *K*5/14, 8/18, 19	**p63**, **ER, FOXA1, GATA3, ZNF217, C/EBPβ, SP1**	[64,65,66,67,68,69]
**Stratified Epithelia Complex**
**Esophagus:***Basal layer: K*5/14, 15. *Differentiated layers: K*4/13	**p63**, **SOX2**, **KLF4**, **KLF5**, **GATA4, ESE-1**	[70,71,72,73,74,75,76,77,78,79]
**Oral mucosa:***Basal layer: K*5/14, 15. *Differentiated layers: K*4/13	**p63**, **SOX2**, **PITX1, AP-2α, c-MYC, GRHL2**	[60,80,81,82,83]
**Cornea:***Basal layer: K*5/14, 12*Differentiated layers: K*3*/12	**p63**, **KLF4**, **KLF6**, **PAX6, FOXC1, ESE-1, RUNX1**	[84,85,86,87,88,89,90,91,92,93,94]
**Skin epidermis:***Basal layer: K*5/14, 15, 19*Differentiated layers: K*1/10, 2, 9, 6 a,b,c*/16, 17	**p63**, **KLF4**, **RUNX1, RUNX3, NF-κB, AP1, AP2, SP1, C/EBP, T3R, RAR, CHOP, GR, SETX, C-MYC, STAT3, NRF2, C/EBPα**	[95,96,97,98,99,100,101,102,103,104,105,106,107,108,109,110,111,112,113,114,115,116,117]
**Hair Follicle Keratin Complex**
**Hair follicle:***ORS: K*5/14, 15, 6 a,b,c*/16, 17, 19*Companion layer: K*6 a,b,c,*/16, 17, 75, 79*IRS: K*25-28/71-74*Hair shaft: K*31-40/81-86	**LEF1, TCF3, SP1, AP2-like, NF1-like, NF-κB, LHX2, STAT3**, **SOX9** **(ORS), HOXC13 (hair shaft), GATA3 (IRS)**	[113,118,119,120,121,122,123,124,125,126,127,128]
**Nail:***Matrix: K*5/14, 1/10, 6 a,b,c*/16, 17, 31, 34, 85, 86*Nail bed: K*6 a,b,c*/16, 75	**GATA3, MSX2, FOXN1, PAI2**	[117,124,129,130,131,132]
**Stress-Induced Keratin Complex**
*K*6 a,b,c*/16,17	**p53**, **STAT1, Gli1/2, Nrf, BARX2, AP1/2, NF-κB, SP1, GR, RAR, RUNX1**	[103,106,109,133,134,135,136,137,138,139,140,141,142,143,144]
Keratins: Type I, Type II	TFs: Kruppel-like, p53 family, SOX family.

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
