# Peer review of "A Kaleidoscope of Keratin Gene Expression and the Mosaic of Its Regulatory Mechanisms"

_ijms, 2023, doi:10.3390/ijms24065603_

Round 1

Reviewer 1 Report

Overall, this is a well-prepared, comprehensive and timely review on principles governing expression of type I and type II keratin loci. It reveals how much has been done by many groups in the field to investigate cell-specific expression of keratins and keratin pairs and how little is really understood. I suggest some additional input to further increase the review's visibility and usefulness to the epithelial biology audience, among others. As the authors point out, there is a wealth of transcription factors, chromatin regulators and super/enhancers, all of which regulate keratin expression. At present, it remains unknown how these different levels of regulation are integrated. Clearly, this is not the authors fault. The biggest question is whether there is a hierarchy in the global regulation of the two keratin gene loci, such that the loci are repressed in non-epithelial tissues and active or activatable in epithelia. What argues against this is that many papers report expression of K8,18 and 19 in non-epithelial tissues /e.g. Muriel et al., 2020; Bader, Jahn, Franke, 1988). Those papers shoud be cited to illustrate extra-epithelial keratin expression in normal tissues. I have the following suggestions:

1. Figure 1: from the way it is presented, it appears that a given keratin, eg K8, is regulated by a different and partially non-overlapping set of transcription factors. This implies that there is no overarching principle. Is this really the case? The data in Fig 1 also raise the question whether published data share any light on the question whether keratin pairs can be regulated in a pairwise manner. That could be addressed in the text. Typo: P63 should read p63.

2. It would be most helpful to include a map of the type I and type II loci presenting the full array of keratin genes plus major transcription factors, enhancer sites and TADs mentioned in the text.

3. There is a number of reports on the regulation of keratins by miRNAs. Does this shed some light on ways by which pairwise expression might be regulated?

4. Line 42, Roth, Friedrich, Biomolec Concepts 2012 should be added.  L103: the mouse genome has non K3 gene, please explain that most of the text refers to human keratin data. L123: a figure illustrating hair keratin expression would help a lot! L131: phenotype is mild to to compensatory upregulation of K19 except in hepatocytes. L125: for a different view, see Cockburn et al., Nat Cell Biol 2023. L136: see also JCI insight Jan 2023 by Engelhardt group. L174: does a comparison of K8 and K18 regulatory elements allow conclusions as to why they are frequently co-regulated but sometimes not? If so, this could be explained. L192 onwards: from here on, they are many problems with grammar, e.g. it reads "regulators of keratins where it should be regulator; or biding instead of binding. It is necessary to perform a thorough spell check on the entire ms.. The legend to Figure 2 does not really explain the figure; it should be improved.

Author Response

Dear Editor,

In the revised version of the manuscript, we addressed all the issues and substantially improved the text, and added two new figures. Our point-to-point replies are followed: 

Point 1. 

The biggest question is whether there is a hierarchy in the global regulation of the two keratin gene loci, such that the loci are repressed in non-epithelial tissues and active or activatable in epithelia.

What argues against this is that many papers report the expression of K8,18 and 19 in non-epithelial tissues /e.g. Muriel et al., 2020; Bader, Jahn, Franke, 1988). Those papers should be cited to illustrate extra-epithelial keratin expression in normal tissues.”

REPLY:

We appreciate the reviewer for this notion. We cited the recommended articles in our revisited version of the manuscript and mentioned the fact of non-epithelial keratin expression in a paragraph concerning simple epithelia keratins. The expression of simple epithelia keratins in non-epithelial tissues is shown at both mRNA and protein levels. During the manuscript preparation, we did not focus on this fact because we were more interested in switching keratin expression in epithelial tissues, where the spectrum of expressed keratins is much wider.

Point 2.

Figure 1: from the way it is presented, it appears that a given keratin, eg K8, is regulated by a different and partially non-overlapping set of transcription factors. This implies that there is no overarching principle. Is this really the case? The data in Fig 1 also raise the question whether published data share any light on the question whether keratin pairs can be regulated in a pairwise manner. That could be addressed in the text. Typo: P63 should read p63.”

REPLY:

Indeed, analysis of published data shows that keratin genes are allegedly regulated by different TFs in different cell types. However, we note that the picture of mechanisms governing keratin gene regulation is obviously incomplete to date. We thus suppose that (i) there is a limited number of “core” TFs required for keratin promoter activation in a cell type-independent manner (for instance, KLF and p53 families), and (ii) in some cell types and/or biological conditions a wide range of additional TFs could be involved in this process. Concerning an “overarching principle”, we suggest that such principle could be manifested in a cooperative action of different TFs binding keratin promoters and regulatory elements. The abovementioned “core” factors potentially open chromatin and establish a platform for the recruitment of other players acting as fine-tuners and required for a flexible modulation of keratin gene transcription in response to internal and external stimuli.

The question about coordinated regulation of paired keratins is a cornerstone one. Notably, there are some other examples of coordinated gene expression (globin genes as a classical model). However, currently available data do not give any tips to solve this problem. Several scenarios to be considered: (i) precise control by trans-acting factors such as TFs and non-coding RNAs; (ii) coordination by spatial interactions within a common “activatory hub” as it has been previously proposed for globin and other erythroid genes; (iii) control at the level of mRNA processing and translation; (iv, intriguing) presence of some control mechanisms based on intranuclear keratins (feedback loop or something like this). From our point of view, discussion of such speculative possibilities requires a separate (mini)review or “opinion”.

Typo is fixed in the revised version.

Point 3.

It would be most helpful to include a map of the type I and type II loci presenting the full array of keratin genes plus major transcription factors, enhancer sites and TADs mentioned in the text.”

REPLY:

Following the reviewer’s suggestion, we prepared this figure for the revised version of the MS. We uploaded the new figure with our reply. 

Point 4.

There is a number of reports on the regulation of keratins by miRNAs. Does this shed some light on ways by which pairwise expression might be regulated?”

REPLY:

Indeed, the role of miRNAs in keratin gene regulation is well-documented. To this end, we suggest that detailed discussion of this aspect of keratin expression is out of the scope of our review. Nonetheless following the reviewer’s suggestion, we added a short citation to the revised version of the MS: “Interestingly, p63 also modulates expression of miRNAs involved in regulation of keratin gene transcription (for the review of the miRNA role in keratin biology see PMID: 32806619).”

Point 5.

Line 42, Roth, Friedrich, Biomolec Concepts 2012 should be added.

REPLY:

Following the reviewer’s suggestions, we cited this article.

Point 6. 

The mouse genome has no K3 gene, please explain that most of the text refers to human keratin data.

REPLY:

We thank the reviewer for this notion. In the revised version of the MS, we clearly state the gene origin, where appropriate.

Point 7.

L123: a figure illustrating hair keratin expression would help a lot!

REPLY:

Following the reviewer’s suggestion, we prepared this figure for the revised version of the MS. We upload the new figure with our reply. 

Point 8.

L131: phenotype is mild to compensatory upregulation of K19 except in hepatocytes.

REPLY:

We thank the reviewer for this notion and pointed out the compensation for the loss of keratin 18 by keratin 19. 

Point 9.

L125: for a different view, see Cockburn et al., Nat Cell Biol 2023.

REPLY:

We appreciated the reviewer for the very interesting recommended article. The appearance of keratin 10 expression in the epidermis before the keratin 1 at a protein level is a novel fact of epidermal differentiation and also indicates the importance of detailed investigation in the field of paired keratin expression activation. We cite the recommended article and emphasize the importance of further studies to determine the principles of paired keratin transcription activation from loci on different chromosomes

Point 10. L136: see also JCI insight Jan 2023 by Engelhardt group.

REPLY:

The recommended article highlights the different functions of keratins 14 and 15 in lung airway epithelia. In this line, we pointed to the ability of keratin 15 to replace keratin 14 in skin stratified epithelia. We clarify this in the paragraph. We also cited the recommended article in a paragraph about pseudostratified epithelia. 

Point 11.

L174: does a comparison of K8 and K18 regulatory elements allow conclusions as to why they are frequently co-regulated but sometimes not? If so, this could be explained.

REPLY:

Similar to other keratin genes, regulatory elements of K8 and K18 are poorly characterized and data on the enhancer landscape within the locus are rather fragmentary. For example, it is unclear whether enhancers of K8 are involved in the regulation of K18, and vice versa. We thus suggest that the available information is insufficient for the conclusions about mechanisms ensuring K8/K18 co-regulation.

Point 12.

L192 onwards: from here on, they are many problems with grammar, e.g. it reads "regulators of keratins where it should be regulator; or biding instead of binding. It is necessary to perform a thorough spell check on the entire ms. 

REPLY:

We thank the reviewer for this notion. The revised version of the MS was carefully proofread.

Point 13. 

The legend to Figure 2 does not really explain the figure; it should be improved.

REPLY:

Following the reviewer’s suggestion, we expand the figure legend as follows: “Schematic representation of the major levels of 3D genome organization. At the level of the whole nucleus and within chromosome territories, active and repressed genome loci are spatially segregated within A- and B-compartments. These structures are formed by large-distant contacts between regions possessing similar profiles of epigenetic marks. Regions forming inactive compartments tend to locate at the nucleus periphery where they establish contact with the nuclear lamina. At the level of 100-1000 kb, chromosomes are folded into globular topologically associated domains (TADs). In mammals, TADs are predominantly formed by cohesin-driven chromatin extrusion frequently terminating at CTCF-binding sites. In a number of well-characterized genome loci, TAD boundaries restrict the areas of enhancer action allowing regulatory systems to activate genes located within a TAD. According to the current paradigm, activation is provided by loop formation between enhancer and cognate promoter.

Best regards,

Ekaterina Kalabusheva

Reviewer 2 Report

The authors have written a good review article on the role of genes in keratin expression and regulation. It is quite detailed and the central building block is Table 1. Nevertheless, there is still room for improvement.

-The figure 2 needs a figure legend for a better understanding.

-The article does not make clear distinctions between human and murine situations, although the genes can be localised at different gene loci. For example, keratin 18 can be found on chromosome 12 in humans and on chromosome 15 in mice. A clearer delineation in the text would have been desirable here. A comparison between the mainly known keratin fibre proteins of humans and mice would also have been very good. Perhaps an illustration would be helpful in this regard.

-The title talks about gene expression and propagates the regulatory mechanism, but unfortunately there is very little in the text about the role of mRNA. What is the situation of the exon/intron structure or slice variants in the keratins mainly mentioned? A more detailed description of this would be very helpful.

-Finally, the chapter on disease caused by mutations in the keratin genes could be given a greater role. As the last chapter with only 7 sentences, I think it is too important and should be expanded somewhat. For example, sebocytomatosis as a mutation of the keratin-17 gene or ichthyosis caused by mutated keratin-1 could be mentioned. This would underline the importance of keratins for health.

Author Response

Dear Editor,

In the revised version of the manuscript, we addressed all the issues and substantially improved the text. Our point-to-point replies are followed: 

Point 1.

The figure 2 needs a figure legend for a better understanding.

REPLY:

Following the reviewer’s suggestion, we expand the figure legend as follows: “Schematic representation of the major levels of 3D genome organization. At the level of the whole nucleus and within chromosome territories, active and repressed genome loci are spatially segregated within A- and B-compartments. These structures are formed by large-distant contacts between regions possessing similar profiles of epigenetic marks. Regions forming inactive compartments tend to locate at the nucleus periphery where they establish contact with the nuclear lamina. At the level of 100-1000 kb, chromosomes are folded into globular topologically associated domains (TADs). In mammals, TADs are predominantly formed by cohesin-driven chromatin extrusion frequently terminating at CTCF-binding sites. In a number of well-characterized genome loci, TAD boundaries restrict the areas of enhancer action allowing regulatory systems to activate genes located within a TAD. According to the current paradigm, activation is provided by loop formation between enhancer and cognate promoter.

Point 2.

-The article does not make clear distinctions between human and murine situations, although the genes can be localised at different gene loci. For example, keratin 18 can be found on chromosome 12 in humans and on chromosome 15 in mice. A clearer delineation in the text would have been desirable here. A comparison between the mainly known keratin fibre proteins of humans and mice would also have been very good. Perhaps an illustration would be helpful in this regard.

REPLY:

We thank the reviewer for this notion. In the revised version of the MS, we clearly state the gene origin, where appropriate.

Point 3.

-The title talks about gene expression and propagates the regulatory mechanism, but unfortunately there is very little in the text about the role of mRNA. What is the situation of the exon/intron structure or slice variants in the keratins mainly mentioned? A more detailed description of this would be very helpful.

REPLY:

The role of alternative mRNA splicing of keratin genes remains poorly understood. Two splice variants of the K80 mRNA with normal and shortened C-terminal domain are expressed in epithelial cells. Noteworthy, the shortened K80.1 variant expression occurs in normal cells and is restricted to several cell types in contrast to the almost ubiquitous K80 expression (PMID: 20843789). In another report the shortened K31 form was described. The shortened form is formed due to polymorphism in 5'-splice site occurring in about 5% of the human population (PMID: 9405442). Mutations of splice sites drive the formation of aberrant splice variants of the K10 mRNA associated with severe hereditary skin disease (PMID: 20302579).

Since the picture of the role of splice variants in keratin regulation is obviously incomplete, this theme requires a separate review. However, taking into account the reviewer’s suggestion, we added the following sentence to the revised version of the MS: Several noncanonical skin or hair keratins could be identified in hair follicles of different mammalian taxons, in particular,  including a unique mammal-specific splice variant of keratin 80 in human (PMID: 20843789) or mucosa-related keratin 4 and 13 in cashmere goats.

Point 4

-Finally, the chapter on disease caused by mutations in the keratin genes could be given a greater role. As the last chapter with only 7 sentences, I think it is too important and should be expanded somewhat. For example, sebocytomatosis as a mutation of the keratin-17 gene or ichthyosis caused by mutated keratin-1 could be mentioned. This would underline the importance of keratins for health.

REPLY:

Following the reviewer's recommendation we expand the paragraph about disorders associated with mutations in keratin genes. In our review, we briefly discuss the keratinopathies to highlight the importance of discovering the regulatory sequences in keratin loci.  Keratin-associated inherited diseases are usually characterized by very similar symptoms caused by several mutations in different keratin genes in a certain epithelial type. Some of these pathologies could be also caused by  mutations in regulatory non-coding regions and changes in  epigenetic profiles disturbing the keratin expression program. We agree that this paragraph should include specific examples of these genetic disorders, thus we added the data about bullosa epidermolysis, pachyonychia congenita, and also epidermolytic ichthyosis.

Best regards,

Ekaterina Kalabusheva

Reviewer 3 Report

The authors provide a concise review of keratin expression and its regulation. Key questions of future studies are defined. The manuscript is largely well written. However, its focus is narrow. Several improvements of the text and addition of more information is suggested.

Line 12: “Keratins are a large superfamily of intermediate filaments” is not correct. Keratins are not filaments but intermediate filament proteins. Is superfamily the right term or is it a family?

“Figure 1” is not a figure, but a table.

Line 26: What are the “Human and Mouse Genome Nomenclature Committees”? The mouse has more type II hair keratins, also K42 (doi: 10.1111/j.0022-202X.2004.22422.x.). The authors should not mix statements about human and mouse. As the mouse is the most important model species in keratin research. The information about mouse keratins should be accurate. Not only the keratin genes are different between species, but also the expression pattern. An example is K2, which is expressed in the skin of all human body parts but only in ear and sole skin of mice (doi: 10.1038/jid.2014.197).

Other species of mammals should be mentioned at least briefly, because there are very informative examples of keratin regulation. For instance, K42 and K124 are specifically expressed in hoof epidermal lamellae of the horse. doi: 10.1371/journal.pone.0219234.

The important aspect of evolution of gene regulation is only briefly touched in line 268 by the statement “evolutionary conserved in related species”. Yet, comparison of gene regulation in different species is likely helpful in understanding the regulation of keratins. Examples include doi: 10.1177/1176934319862246

Line 70: “Simple epithelia keratin complex (KC)” is a term not regularly used in the literature on keratins. KC is widely used as abbreviation for “keratinocyte”. A new meaning of “KC” is confusing. It is not clear what “complex” means here. A different word should be used, or just “Simple epithelia keratins”. Moreover, in line 297 the epidermal differentiation complex (EDC) refers to a complex of physically linked genes, whereas the authors talk about a group of type I and II keratins at different loci.

Line 104 and following: K6 isoforms KRT6A, KRT6B, KRT6C should be mentioned.

Line 121: where are keratins K39 and K40 expressed?

Line 307: “proteins called β-keratins” is not inaccurate and probably very confusing to the reader, because these proteins are very different from the topic of this article, that is, (true) keratins. “proteins called β-keratins” are definitely NOT keratins (intermediate filament proteins). This fact should be stated clearly.

Lines 324-326: References are needed here.

In general, the manuscript would benefit from adding more references.

Important recent papers should be cited and discussed, e.g. doi: 10.1126/sciadv.abo7605.

Author Response

Dear Editor,

In the revised version of the manuscript, we addressed all the issues and substantially improved the text, and added new figures. Our point-to-point replies are followed: 

Point 1 

Line 12: “Keratins are a large superfamily of intermediate filaments” is not correct. Keratins are not filaments but intermediate filament proteins. Is superfamily the right term or is it a family?

REPLY

We appreciate the reviewer for pointing out our inaccuracy. We correct the sentence in the following manner: “Keratins are a family of intermediate filament-forming proteins highly specific for epithelial cells”

Point 2 

“Figure 1” is not a figure, but a table.

REPLY:

Following the reviewer’s suggestion we made the correction. 

Point 3

Line 26: What are the “Human and Mouse Genome Nomenclature Committees”? The mouse has more type II hair keratins, also K42 (doi: 10.1111/j.0022-202X.2004.22422.x.). The authors should not mix statements about human and mouse. As the mouse is the most important model species in keratin research. The information about mouse keratins should be accurate. Not only the keratin genes are different between species, but also the expression pattern. An example is K2, which is expressed in the skin of all human body parts but only in ear and sole skin of mice (doi: 10.1038/jid.2014.197).

REPLY:

We thank the reviewer for this notion. In the revised version of the MS, we clearly state the gene origin, where appropriate.

Point 4

Other species of mammals should be mentioned at least briefly, because there are very informative examples of keratin regulation. For instance, K42 and K124 are specifically expressed in the hoof epidermal lamellae of the horse. doi: 10.1371/journal.pone.0219234.

REPLY:

We appreciate the reviewer for the recommendations. We cite the above mentioned article and also add more details about mouse and cashmere goat keratin expression patterns. 

Point 5

The important aspect of evolution of gene regulation is only briefly touched in line 268 by the statement “evolutionary conserved in related species”. Yet, comparison of gene regulation in different species is likely helpful in understanding the regulation of keratins. Examples include doi: 10.1177/1176934319862246.

REPLY:

We pay the Reviewer’s attention to the fact that in line 268 (and in the entire paragraph) we do not discuss evolutionary aspects of gene regulation. Here, we briefly describe features of TAD boundaries in mammalian genomes and note that the TAD boundary profile is largely conserved in cell types and between related species. TAD boundaries restrict areas of enhancer action at least in some well-described “model” loci and thus may dictate evolutionary conserved regulatory landscapes. However, this is extensively discussed in a number of reviews, for example see https://doi.org/10.1016/j.gde.2020.12.015

Point 6

Line 70: “Simple epithelia keratin complex (KC)” is a term not regularly used in the literature on keratins. KC is widely used as an abbreviation for “keratinocyte”. A new meaning of “KC” is confusing. It is not clear what “complex” means here. A different word should be used, or just “Simple epithelia keratins”. Moreover, in line 297 the epidermal differentiation complex (EDC) refers to a complex of physically linked genes, whereas the authors talk about a group of type I and II keratins at different loci.

REPLY:

 We thank the reviewer for this notion. We removed the abbreviature “KC” to avoid misunderstanding. We introduced the concept of keratin complex for a more detailed description of keratin expression patterns related to certain types of epithelia or pathological conditions. We were aimed to point out that keratin expression could be characterized not only as paired (which implies the expression control of single genes from different chromosomes), but also as “complex” ensured by  coordinated transcription of several genes from different loci. 

Point 7.

Line 104 and following: K6 isoforms KRT6A, KRT6B, KRT6C should be mentioned.

REPLY:

In the revised version of the MS, we briefly highlighted expression patterns of K6 isoforms in a table including data about keratin expression patterns. 

Point 8

Line 121: where are keratins K39 and K40 expressed?

REPLY:

Keratin 39 was found in the upper hair cuticle and the upper cortex, while keratin 40 is expressed in the upper hair cuticle of human hair follicles according to Langbein et al, 2007(PMID: 17301834). We prepared the figure describing the hair follicle keratin expression pattern (new figure X in the revised MS). 

Point 9

Line 307: “proteins called β-keratins” is not inaccurate and probably very confusing to the reader, because these proteins are very different from the topic of this article, that is, (true) keratins. “proteins called β-keratins” are definitely NOT keratins (intermediate filament proteins). This fact should be stated clearly.

REPLY:

We modified the sentence as follows: “Finally, there is a noteworthy example of the role of the 3D genome organization in providing regional specification of skin appendages of birds. In addition to conventional α-keratin type I and II, birds and reptiles have a special class of intermediate filament proteins (β-keratins or corneous β-proteins, while structurally these proteins are not related to keratins), which serve as “building blocks” of various skin derivatives.

Point 10

Lines 324-326: References are needed here.

REPLY:

Following the reviewer’s notion, we added an appropriate citation: PMID: 34208174 

Point 11

In general, the manuscript would benefit from adding more references.

Important recent papers should be cited and discussed, e.g. doi: 10.1126/sciadv.abo7605.

REPLY:

In the revised version of the MS, we added the following paragraph: Recently, it has been shown that such distant contacts within mouse keratin locus are regulated by DNA deoxygenases Tet2/3 in DNA methylation-dependent and DNA methylation-independent manner (PMID: 36630508). These data are of particular interest, because it raises the question about potential modulation of keratin gene expression by artificial recruitment of DNA-methyltransferases (PMID: 27662091).”

Best regards,

Ekaterina Kalabusheva

Reviewer 4 Report

In this Review the authors describe pattern of keratin expression and the underlying mechanism of regulation.

I am not an expert in the field of keratin expression. Some things may be clearer for people who have many years of experience in the field than for me. I did not understand on which basis the keratins covered in this review were selected. I also found the organization of the paragraphs a little confusing. E.g., the first paragraphs follow table 1 but then there a two additional are added (Hair KC and Typ II keratins). I am missing the consistency. Why is there no paragraph for Typ I?

It should be clearer stated what are mouse date and what are human date. It might be clear that E 15.5 refers to mouse, but it would be good to name it.

Line 140: only enhancer?

Author Response

Dear Editor,

In the revised version of the manuscript, we addressed all the issues and substantially improved the text. Our point-to-point replies are followed: 

Point 1.

I did not understand on which basis the keratins covered in this review were selected.

REPLY:

Keratin genes belong to a limited cohort of genome loci containing a large number of ortholog genes. A complex genome environment suggests the presence of regulatory systems discriminating and selectively activating certain genes in a tissue- and cell type-specific manner. These systems mainly based on spatial interactions between locus control regions and promoters have been partially described in globin and PCDH loci. However, little is known about precise regulation of keratin expression at the level of transcription factor networks and chromatin 3D structure. This made us summarize the available data in the field in order to understand possible directions for future research.

Point 2. 

I also found the organization of the paragraphs a little confusing. E.g., the first paragraphs follow table 1 but then there a two additional are added (Hair KC and Typ II keratins). I am missing the consistency. Why is there no paragraph for Typ I?

REPLY:

We appreciate the reviewer for pointing out the problems with the manuscript paragraph organization. This misunderstanding appeared due to the table separate the chapter describing the keratin expression pattern. We take into account that the table position disturbs the reading and move it at the end of the chapter. 

Point 3.

It should be clearer stated what are mouse data and what are human data. It might be clear that E 15.5 refers to mouse, but it would be good to name it.

REPLY:

We thank the reviewer for this notion. In the revised version of the MS, we clearly state the gene origin, where appropriate.

Point 4. 

Line 140: only enhancer?

REPLY:

To clarify this statement, we changed this as follows: “In higher eukaryotes, enhancer elements containing clustered binding sites for transcription factors are the main regulators of the tissue-specific gene expression”.

Best regards,

Ekaterina Kalabusheva

Round 2

Reviewer 2 Report

The authors answered all my concerns and comments very well and now the manuscript is in good acceptable condition. Thanks very much.

Reviewer 3 Report

The manuscript has been substantially improved by changes of the text and by adding figures 1 and 2. There is one erroneous word which should be removed when the authors correct the proof: In line 363 (... a special class of intermediate filament proteins...) please delete "intermediate", because beta-keratin filaments have a diameter of 3.4 nm which is not intermediate between actin microfilaments (7 nm) and microtubules (25 nm). The manuscript reads very well.

Reviewer 4 Report

The manuscript has benefited from the adjustments and is now clearer and more structured. In this form, I can favor a publication.